# Medical Augmented Reality: Definition, Principle Components, Domain Modeling, and Design-Development-Validation Process

**DOI:** 10.3390/jimaging9010004

**Published:** 2022-12-23

**Authors:** Nassir Navab, Alejandro Martin-Gomez, Matthias Seibold, Michael Sommersperger, Tianyu Song, Alexander Winkler, Kevin Yu, Ulrich Eck

**Affiliations:** 1Computer Aided Medical Procedures & Augmented Reality, Technical University Munich, DE-85748 Garching, Germany; 2Laboratory for Computational Sensing and Robotics, Johns Hopkins University, Baltimore, MD 21218, USA; 3Research in Orthopedic Computer Science, Balgrist University Hospital, University of Zurich, CH-8008 Zurich, Switzerland; 4Department of General, Visceral, and Transplant Surgery, Ludwig-Maximilians-University Hospital, DE-80336 Munich, Germany; 5medPhoton GmbH, AT-5020 Salzburg, Austria

**Keywords:** Augmented Reality, Medical Augmented Reality, surgical data science, Artificial Intelligence, multi-modal sensing, computer vision, acoustic sensing, perceptual visualization, sonification, validation

## Abstract

Three decades after the first set of work on Medical Augmented Reality (MAR) was presented to the international community, and ten years after the deployment of the first MAR solutions into operating rooms, its exact definition, basic components, systematic design, and validation still lack a detailed discussion. This paper defines the basic components of any Augmented Reality (AR) solution and extends them to exemplary Medical Augmented Reality Systems (MARS). We use some of the original MARS applications developed at the Chair for Computer Aided Medical Procedures and deployed into medical schools for teaching anatomy and into operating rooms for telemedicine and surgical guidance throughout the last decades to identify the corresponding basic components. In this regard, the paper is not discussing all past or existing solutions but only aims at defining the principle components and discussing the particular domain modeling for MAR and its design-development-validation process, and providing exemplary cases through the past in-house developments of such solutions.

## 1. Introduction

Probably the first construction of an Augmented Reality (AR) system goes back to Filippo Brunelleschi around 1413 [1]. Following concepts developed by Ibn al Haytham, known in the west as Alhazen or ’the father of modern optics’ [2,3], Brunelleschi created a drawing of a cathedral from a given point of view according to his linear perspective method. To confirm the correctness of his perspective projection, he drilled a hole into the drawing at the position of the observer’s eye and positioned a mirror in front of the hole. The observer could move the mirror and clearly see the matching view of the physical world and the virtual representation. He went on to cover the parts on top of the building he had drawn in silver coating so that the observer would see the actual sky double reflected onto the mirror to improve the realism of the AR scene with the current sky on top of the drawn virtual building [4], as shown in Figure 1 (We strongly encourage the reader to visit https://www.youtube.com/watch?v=G2BCdA23Kpg (accessed on 12 December 2022) to observe the nicely made video, visualizing Brunelleschi’s concept). The reflected sky made the augmentation more natural and provided a convincing perceptual rendering of the augmented scene. It is fascinating to see that the components of this Renaissance-era system designed, built, and validated by Brunelleschi closely match the modern ones presented here.

While many definitions of AR have been given throughout the last decades, we will try to provide one inspired by all these definitions and try not only to consider the impact of novel technologies and concepts but also bring in basic philosophical discussions about the current and future role of AR in education, training, and performance of a variety of personal and professional tasks. We furthermore point the reader to the literature review about Medical Augmented Reality (MAR) by Sielhorst et al. from 2008 [5] and a more recent review about the use of head-mounted displays (HMD) in surgery by Birlo et al. [6]. Furthermore, an example of adopting one of the most used commercially available HMD systems is given in a review paper by Gsaxner et al. [7].

One of the most commonly applied definitions is the one proposed by Ron Azuma in 1997 [8]. Most definitions consider AR as the addition or integration of virtual elements within the real world perceived by the user. The perception could follow all human senses, including sight (vision), auditory (hearing), smell (olfaction), taste (gustation), and touch (tactile perception). An AR system may augment one or multiple of these senses. The most important outcome of such augmentation is to influence the mental mapping of the users and to enable a higher level of learning and decision-making experiences. One major role of AR is to increase the knowledge and awareness of users about their environment and the subjects they interact with. To define and describe the basic components of any AR solution, we need to realize that for an AR system to augment the environment and present it to the users, it needs to understand and model its environment and consider the objectives and perceptual capabilities of its users. Therefore, we define the following components as essential for the design and development of AR systems:1Physical-World Modeling2Display, including Augmentation and Perceptual Rendering3Interaction4Evaluation, including functional Testing & Validation as well as Ethics

Let us focus on the role MAR could or should play in our society, particularly in healthcare-related fields. Due to the recent commercial availability of AR display systems, many clinical, academic, and industrial labs have focused on low-hanging fruits aiming at the superimposition of virtual to real objects, often for navigation purposes. We believe, however, that the community will move to more intelligent AR solutions, enabling higher levels of understanding and mental mapping rather than the direct superimposition of an insertion path or a deep-seated target onto the expert’s view of the surgical scene. Since the time of Aristotle, we know that humans use all of their senses to explore the environment. Humans interact with their environment and remember experiences, which gradually turn into intuitions: When you look at a wine glass, you know that it can break. None of your senses are able to detect this, but you infer this familiarity from past observations and experiences. These experiences are, in turn, based on your human senses. AR could allow us to make experiences based on much more powerful sensing than our limited human sensing capabilities, accumulate greater knowledge, and let it turn into intuitions. The role of AR, therefore, becomes at least twofold: On one hand, it fetches and brings relevant information to our direct attention. On the other hand, AR allows us to build a deeper understanding of our environment to perform better, even without direct augmentation, as we have learned and created a better understanding. This shows what an important tool MAR can be in teaching and training not only for clinicians and clinical staff but also for the population who strives for better and more in-depth knowledge of their own anatomy and physiology and the changes it goes through during our lifetime.

In addition to the multi-modal sensing and digital reconstruction of the physical world and its perceptual augmentation, a major component of any Medical Augmented Reality System (MARS) is its user-centered, dynamic, and continuous evaluation. Since AR augments human actors, its design, development, and deployment must include constant and dynamic user evaluation. The best MARS can be developed not only after in-depth use case studies but also through the dynamic participation of users in each step of its iterative design and development. It is important to notice that scientists or clinicians can only evaluate one particularly designed solution. It is generally impossible to evaluate AR as a whole. Therefore, one has to be very careful about publications claiming to “evaluate AR” or its effect on users while they are only evaluating their particular system and methodology.

Finally, ethics needs to be considered from the very first moments of problem definition throughout the iterative design, development, evaluation, and deployment process. This is crucial in particular for MARS, in which not only the well-being of an individual patient is at stake, but potentially the healthcare of human society and its future development.

In the early 1990s, Bajura et al. [9] presented one of the first MARS, superimposing ultrasound images onto a pregnant volunteer. In the process, they identified the problem of misleading depth perception when virtual anatomy naively occludes the patient. To address this issue and to make the viewer perceive the anatomy to be located within the target anatomy instead of in front of it, they render a “synthetic hole around ultrasound images in an attempt to avoid conflicting visual cues”. It is astonishing what they presented at the time and is humbling to a current researcher to see that MAR applications and their issues from the early 90s could be explained with the same system model and implicitly studied and worked on similar basic concepts but with limited power of computation and at a time when computer vision and medical imaging were not yet taking advantage of powerful machine learning methodologies.

In the following sections, we introduce the proposed Medical Augmented Reality Framework, give a detailed description of all of its components, and showcase exemplary MARS developed by the Chair for Computer Aided Medical Procedures and its clinical, academic and industrial partners, and discuss their basic components in light of the framework.

## 2. The Medical Augmented Reality Framework

To define MAR and enable the systematic design and validation of these systems, we propose a framework as illustrated in Figure 2. An MARS is composed of three main components: (i) the *Digital World*, (ii) the *AR/VR Display*, and the (iii) *AR/VR User Interaction* component. The system interacts with the *Physical World* by creating a digital representation of its environment and interacting with the user through a dynamic UI. As an overarching methodology, a dynamic and continuous *Evaluation* forms an essential part of the conceptualization, development, and translation of an MARS, covering aspects from system validation to ethics and regulatory matters. In the following sections, we will explain all components of this framework in detail.

### 2.1. Physical World

AR is characterized by enabling the co-localization and display of digital and physical information in the physical space, realized through key technologies such as spatial computing. The physical world is not a component of an MARS. Still, the system needs to sense and understand its environment to have both content related to the physical world and a canvas to represent it.The users are part of the physical world, and the focus of the combination of real and virtual content, so that they perceive the augmentations through their own senses.

In a medical environment, the physical world comes with particular challenges.The operating room (OR) is a highly dynamic and crowded environment with high-intensity lighting conditions which, for example, influence the inside-out tracking systems of AR-enabled devices. Furthermore, additional sensing devices can capture aspects of the physical world that humans cannot perceive, e.g., the internals of a human body, using medical imaging or additional sensor data from other external medical equipment.

### 2.2. Digital World

For an AR system to create a meaningful augmentation of the physical world, it must be able to reconstruct and interpret it digitally. Several companies in the AR field express the goal of creating digital twins of the environment, e.g., entire cities or even another layer of the universe utilizing the enriching interconnectedness of the internet [10]. An AR system, however, does not need to create an exact or complete replication of the environment, but only a good enough understanding for the intended application: An HMD application to superimpose data might need to know the location of the user’s head in a room. A surgical AR navigation application might need a Computed Tomography (CT) or Magnetic Resonance Imaging (MRI) scan of the patient. An intraoperative context-aware AR user interface might need to understand the surgical workflow and means to recognize the ongoing procedural step. Future MARS do not only need to build a digital representation of their environment but also to understand, model, and monitor the role of each of the components acting in such environments such as patient monitoring systems, imaging solutions, surgical robots, instruments, and most importantly, the human actors [11].

Commercial AR-enabled devices today were often designed for the mass market and, therefore, usually include applications that are companions to the user in several activities: Users can access their emails, be notified about appointments in their calendars, have a voice-controlled assistant obtain information from the internet, or stream videos into the user’s AR environment. An MARS, in particular, if used for dynamic clinical decision-making, not only does not require some of these applications but may even need to prevent their installation due to data protection requirements (e.g., a separate network for patient information) and patient safety (avoid distractions of the surgical staff). Clinical devices are often optimized for their consistent and continuous use within medical procedures and are solely used to improve clinical decision-making and healthcare delivery. MAR solutions, which are not made for teaching, training, or patient rehabilitation, but are used within routine clinical procedures, need to focus only on improving them.

#### 2.2.1. Sensors

Like its users, the technical system perceives its environment through several sensors to create an internal representation, which is separate from the physical environment but shaped by the gathered information. Such sensors, e.g., optical, acoustic, haptic, or inertial sensors, can acquire similar information as human senses do. However, they can also observe the environment in more advanced forms than human sensing, translating originally unavailable information for humans to sense and observe beyond their natural abilities. CT, MRI, ultrasound, X-ray, OCT, fluorescence, and hyper-spectral imaging are a few of such sensors used in the medical domain.

The data of the sensors must be calibrated and synchronized. The fused data from the sensors must then be organized before they can be interpreted. Thanks to advancements in computational power and intelligent analysis of the available data, the digital representations of the patient and environment can revolutionize all divisions of the medical domain, from teaching and training to diagnosis, treatment, and follow-ups, to patient information, preventive actions, and rehabilitation.

#### 2.2.2. Perception

The sensor data need to be intelligently processed to enable the digital system to interpret the collected input. In the case of marker-based AR, the system mainly perceives the identity and pose of markers positioned within the environment allowing relative positioning of virtual objects. Feature-based systems enable tracking and localization using natural features. However, the virtual objects were often placed at predefined locations, defined by the AR system designers, relative to the natural feature. Machine Learning and Artificial Intelligence start enabling semantic understanding of scenes, their components, and their interaction within the environment. This could allow AR systems to have higher levels of perceptual understanding of the environment they act in. In some of the most recent work in surgical scene understanding, Ozsoy et al. [11,12] used multiple camera views of surgical scenes to not only build a 4D reconstruction of the scene but also generate semantic scene graphs recovering, modeling and representing complex interaction between surgical staff, patient, device, and tools within the OR. Such high-level computer perception will allow for intelligent workflow-driven solutions [13]. Both marker-based and semantic understanding will continue to be part of AR systems. While Machine Learning and Artificial Intelligence can infer a much broader understanding of the situation in the OR, which the AR system can then augment, marker-based tracking with predefined virtual objects will continue to provide more accurate pose information. This may not be a limitation at all, as most commercial surgical navigation systems rely on some form of markers anyways [14], which the AR system can also rely on.

#### 2.2.3. Digital World

The above processes lead to a digital representation of the environment, which we call the *Digital World*. Here objects of the physical world relate to their digital counterparts. An AR architect can author specific virtual objects with complex behaviors within this digital world. We can, for example, specify a particular virtual object to be attached to a physical one or define behaviors of virtual objects based on changes in the physical world.

Advances in computer vision and Machine Learning, as discussed in the previous section, give us the hope that in the near future, semantic understanding of the surgical environment, its main components, their roles, and interactions may allow for a higher level of AR design and system commands, e.g., visualize and augment the endoscopic view of surgery on the trainee’s headset only during coagulation performed by the head surgeon.

### 2.3. AR/VR Display

The mere modeling and reconstruction of a digital world based on the physical world does not yet lead to an AR system. An AR system must convert this internal representation into messages of valuable augmentations that it can convey to its user. This is achieved by rendering the augmented digital world in a way its users understand. Rendering and displaying in AR are not restricted to creating images displayed on a screen. Besides visual rendering, one can also include audio or haptic rendering. Some of the creative participants of the Medical Augmented Reality Summer School 2021 even proposed to use gustatory sensing to pass pertinent messages to surgeons during minimally invasive surgery (https://www.medicalaugmentedreality.org/mar2021.html, accessed on 1 November 2022).

A good AR rendering is not the mere introduction of virtual data into the OR or an overlay of virtual content onto a real object. It integrates virtual content into the users’ perception of their physical environment to make it a compelling part of their world. For example, when the users of an MARS observe the internal organs not usually visible to their eyes or hear the augmented sound of an organ touched by a surgical instrument, a *perceptually consistent* experience should be preferred, allowing the users to recognize and localize the objects otherwise invisible to the users’ unaided senses. Evidence that they can, in fact, integrate the additional information and are affected by the augmented visuals and sounds can be explored in experiments with users: For example, Okur et al. [15] analyzed the surgeons’ visualization preference (either AR or VR) in intra-operative SPECT acquisitions. The surgeons relied on both AR and VR visualizations and stated that the two modes complemented each other. In an auditive AR experiment, Matinfar et al. [16] had users recognize relevant sensor changes in retinal surgery. The users could accurately and quickly recognize the position of the tool.

In contrast, a large body of work in computer graphics seeks to render images of virtual scenes that appear photo-realistic to a viewer. Including such photo-realistic visualizations of virtual objects in MAR applications might be tempting. However, the creators of these visualizations must decide if they want the user not to be able to tell the physical and virtual objects apart. Some applications may emphasize that an object is virtual and is, therefore, not subject to some behaviors common to the physical world. If a virtual object, e.g., violates occlusions, i.e., is visible below a real surface, a photo-realistic visualization alone would not solve the perceptual issue.

There were several examples of such perceptual visualization projects in the last two decades, of which we summarize a few briefly: Bichlmaier et al. [17] applied a technique called Focus and Context (F+C) rendering of segmented CT scans superimposed on patients on an HMD. They modulated the transparency of the context layer based on the curvature of the surface, the angle between the incidence ray and the surface, and the distance to a focus spot, resulting in a gradual fade out and a “ghosting” effect. Kutter et al. [18] replaced the segmentation of the CT by direct volume rendering and color-filtered the surgeon’s hands to avoid incorrect occlusions between the surgeon’s hands and the AR scene. Martin et al. [19] decomposed different F+C visualization aspects in a comparison of depth perception on HMDs to find the best-performing combinations. They found that hatching and chromatic shadows for the background could effectively improve in-situ visualization. Kalia et al. [20] present the inclusion of interactive depth of focus as a concept to enhance depth perception in AR microscopy. Users reported a better sense of depth if they were given the ability to control the focus distance. In another work, Kalia et al. [21] analyzed the effect of motion parallax for intraoperative AR visualization, in which the scene moved arbitrarily over time. They found significant differences between Mono and Stereo visualizations with motion parallax and without.

While displaying low-dimensional but helpful parameters such as instrument distance or pose information directly to the surgeons can be distracting and unintuitive, leveraging them as a means to generate or modify sound as a form of auditive AR has shown great potential. Matinfar et al. [16] presented “Surgical Soundtracks” augmenting a known piece of music based on parameters extracted from surgical sensing in Ophthalmology to aid contextual understanding and situational awareness. In another Ophthalmology application of auditive AR, Roodaki et al. [22] use physical modeling sound synthesis derived from visual information to improve performance in focus-demanding alignment tasks. Ostler et al. [23] investigated if differences in audio signals acquired from a microphone inside the patient in laparoscopic surgery are detectable merely by listening and found that surgeons gained additional information from the audio signals. Additionally, they could automatically classify instrument–tissue interaction sounds using a Deep Learning method with high accuracy.

### 2.4. AR/VR User Interaction

User interfaces (UI) to interact with machines had already been designed before the concept of user interaction was formalized. Historically, with every advance in computing, new display forms, new input methods, and new interactions were introduced. We saw decidedly different classes of interaction methods coming to fruition roughly every 20 years: Command-line UIs were the dominant method of user interaction in the 1960s and 1970s, where users had to type explicit commands on a keyboard. In the 1980s, computers became more user-friendly with graphical UIs and mouse input devices introduced to the general public. UIs continued to evolve in the 2000s when mobile screens popularized multi-touch controls. In the 2020s, the emergence of commercially available AR systems led to limited commercial deployment of AR solutions for various applications, including industrial maintenance, education, and healthcare.

Established input devices do not seem to be suitable for most AR applications. The community needs to take advantage of rich sensing and advanced algorithms and develop novel user interaction paradigms to make interacting with such systems more natural and context-driven. E.g., multi-modal hand gesture input, eye or head gaze, voice commands, body motion, and AR controllers could serve as more instinctual interactions. While promising implementations have been presented, effective natural interaction solutions are still to be presented and turned into novel standards.

User interaction in the medical domain comes with even more challenges, such as sterility requirements in the OR, technical regulations for incorporating exercises for patient rehabilitation, or the need for high accuracy, e.g., when augmenting precision surgeries. Novel and efficient user interaction paradigms for MAR solutions, particularly in a surgical environment, are still to be developed.

Compelling AR interfaces need to be: *adaptive*, *believable*, and *consistent*: *Adaptive*: AR interactions become instinctual when they are coupled with the context of the user’s environment, task, action, and mental state. Context is not always known at design time. Therefore, UIs need to be adaptive [24]. In AR-assisted medical procedures, an example could be showing different augmentations based on the step in the surgical workflow and the instrument’s position. *Believable*: Physically-based rendering greatly impacts perceived realism. However, the augmented content does not always need to be realistic. In some applications, it may be necessary for the user to distinguish between the real world and its augmentation. In such cases, too high realism could confuse what is real and what is not. *Consistent*: With imprecise inputs from human motion (gestures, gaze, and voice), controllers, and the environment condition, captured by the multi-modal sensors, a computationally driven model is required to process the rich sensing data and turn them into consistent feedback within the AR interface. Once again, novel interaction paradigms may be required to take full advantage of mixed reality. The interactive virtual mirror paradigm we designed and developed is one of the unique interaction solutions explicitly developed for MARS [25,26,27].

### 2.5. Evaluation

The users are the focus of all AR systems and should be considered accordingly during conception and development as it is their perceptual system that receives virtual content from the AR system. With a successful AR system, users can use this information to improve their interaction with their natural environment. One example of the iterative inclusion of the user towards the development of a successful system is a project conducted within our lab that integrated VR and AR for intraoperative single-photon emission computerized tomography (SPECT), which was started in 2007 [28] and resulted in the first MARS used in surgery with CE certification and FDA clearance in 2011 [15].

Hence, acquiring and evaluating usability-related measurements within an AR system is essential in the development process. This observation holds even more true for critical applications such as medical treatment and surgical interventions. Regarding the visual fidelity of AR systems, consideration of human perception and intuitive behavior are essential to enabling immersive and user-friendly experiences. Since intuition may differ widely between humans (even more so when they come from different areas of expertise), designers of AR systems may often take different approaches to solve problems than the ones preferred by the final users, particularly in highly-specialized fields. Introducing methods, paradigms, interactions, novel renderings, or displays that contradict the final users’ intuition may entail an adaptation process to redefine what the users perceive as familiar. Therefore, intermediate and iterative validations involving the final users should be conducted in parallel to the conception and development phases of MARS and not only in the early or final stages.

The integration of user studies during the design and development stages allows for iterative refinement of the systems. Further, it provides insight into the feasibility of integrating AR into medical procedures. These studies should be carefully designed, conducted, and analyzed. Otherwise, they can produce misleading results, wrongly reject valuable concepts, or lead to wrong conclusions regarding how these systems will be adopted by the end-users [29]. For this reason, different stages during the development of MARS should consider distinctive evaluation frameworks. Such frameworks should simultaneously consider the target application of the system and the end-user and must ensure the collection of complete and appropriate qualitative and quantitative data. Examples of these evaluation frameworks can include the use of simulators and phantoms for the development of training systems; or the use of cadaver trials for surgical navigation purposes, replicating the characteristics of the real use case more reliably. In recent years, Machine Learning enabled the in silico generation of realistic data. We believe that such techniques will be more and more used in the early phases of research and development to test and validate ideas. However, the key is to enable both qualitative and quantitative measures of human performance within medical mixed-reality environments.

The acquisition of qualitative measures allows for investigating the systems’ usability, including attributes such as learnability, efficiency, intuitiveness, and/or satisfaction. In addition to collecting experiential feedback from individual medical partners, using surveys or questionnaires is a common approach for gathering qualitative measures and user feedback. The content of the questionnaire depends on the type of study evaluated. Examples of such may include general surveys to gather usability (System Usability Score [30]), mental workload (NASA Task load [31], Surgery Task Load [32]), and cybersickness scores. More specialized qualitative measures may resort to established, but not limited to, questionnaires within the AR/VR community such as those designed for immersion in VR [33], communication and presence [34,35], or immersion in AR [36]. Such qualitative measures, however, are often not as reliable as quantitative measurements of performance. To provide a more objective measurement of computational systems on humans, the affective computing community has been actively creating and evaluating new ways of quantifying human affects. By applying a learning-based algorithm, multi-modal digital data that are collected by wearable sensors can report and inform the physical and mental fatigue levels [37]. We therefore highly recommended thinking of alternative ways for quantitative measurements of human performance, indicating the success of mental mapping or measuring fatigue and/or dissatisfaction.

In addition to the qualitative data, the collection of quantitative data can contribute to investigating the feasibility of integrating the systems during the performance of medical tasks. Such data can determine the system’s accuracy, precision, repeatability, latency, reliability, and stability; and also play a determinant role during the certification process of the system. Quantitative results can be used to investigate if an MARS contributes to reducing the frequency or magnitude of errors, enhancing dexterity, or minimizing the time required to complete a task. These measurements also contribute to understanding how the user benefits from interacting with the system, comparing the efficacy of novel interaction paradigms, or studying how these systems can support collaboration, communication, or task performance in the OR. Therefore, the evaluation frameworks should be designed to produce reliable and repeatable data that reflect the users’ performance while interacting with the system.

Once again, ethics needs to be considered from the very first moment of problem definition and throughout the iterative design-development-validation process. The final solutions of MAR need to be accessible to and affordable for all subjects whose data contribute to the design and validation of such solutions. MAR needs to target the well-being of all of society, particularly when public funds sponsor such research. It is highly recommendable to have experts of ethics personally participate in all MAR projects as team members and for all researchers involved in MAR to take explicit training on ethics of design and usability studies.

## 3. Exemplary Applications of the Medical Augmented Reality Framework

The following sections show selected systems developed over two decades at the Chair of Computer Aided Medical Procedures, for which we dissect their components and map them into the framework. We argue that the proposed MAR framework can describe all MARS from the past, present, and future.

### 3.1. 3D Telepresence Based on Real-Time Point Cloud Transmission

3D Telepresence allows users to collaborate over distances, for example, by sharing a physical space as a real-time reconstruction with remote users connected via secure networks. Remote users may virtually join local users through a VR headset, can communicate through verbal and non-verbal cues, and interact with each other through AR and VR.

As a medical use case, we identified the need for 3D teleconsultation during the first response for accidents and emergencies. E.g., in the case of massive casualties [38] or rare conditions, on-site medical staff often wish to consult with experts to apply the proper treatment. In addition, with telepresence, remote experts are no longer required to travel to local sites; therefore, they are more efficient in switching between cases and avoid unnecessary patient contact [39].

We present *Augmented Reality Teleconsultation for Medicine (ArTekMed)* that fully utilizes the proposed MAR framework, as seen in Figure 3. ArTekMed captures the physical environment of the patient, local paramedics, and surroundings using color and depth (RGB-D) sensors. The system transforms the acquired data into a complete 3D representation of the scene. In this digital world, users may freely choose a virtual viewpoint to inspect the scene. ArTekMed automatically calibrates users wearing optical see-through displays into the relative coordinate systems of the nearby RGB-D cameras [40]. As ArTekMed locates both AR and VR users in the same relative digital space, users can utilize synchronized virtual tools such as 3D drawings and objects to aid communication. Virtual avatars transfer gestures and social presence between the participating users [41].

Visualization and interaction paradigms that are unique to shared virtual spaces (seen in Figure 4), such as magnifying dioramas (Magnorama), aid the exploration of the 3D scene and help users create precise annotations [42]. These shared UIs are visible and interactable to all parties and can aid visualization and verbal communication between remote and local users and co-located users [43]. Further, Projective Bisector Mirrors [44] allow VR users to observe the patient through the eyes of the original high-detail camera view rather than the reconstruction, visualized as a spatially coherent mirror representation.

From our description, we map the components of ArTekMed to our MAR framework:**Physical World of ArTekMed:** The physical world includes the patient with the surrounding environment and objects, comprising local users such as paramedics and bystanders.**Computer Sensors of ArTekMed:** On the patient’s side, the RGB-D cameras, the built-in sensors of the AR-HMD, and microphones. On the remote user’s side, infrared-based sensors capture the LED constellation on the VR headset, controllers, and body tracking pugs.**Computer Perception of ArTekMed:** The computation units connected to the sensors interpret the acquired data. In particular, the inside-out sensors on the AR headset use simultaneous localization and mapping (SLAM) to compute the local user’s first-person perspective correctly. Furthermore, on the VR side, the tracking system returns the six degrees of freedom pose within the digital world.**Digital World of ArTekMed:** The digital world consists of the reconstructed point cloud and virtual avatars to represent the users for conveying non-verbal communication and social presence. Additional tools to aid the consultation, including 3D annotations and the Magnorama, are part of this digital world and are rendered to all participating users.**Rendering Displays of ArTekMed:** The local users wear AR-HMDs for in-situ augmentations inside the real environment. They allow the local user to perceive the real world alongside digital visual and auditive augmentations that ArTekMed generates from co-located and remote-connected participants. In addition, the VR users perceive the reconstructed point cloud within their VR headsets and can talk and listen to the local AR users.**Augmentations of ArTekMed:** Augmented components are virtual elements such as 3D annotations, the Magnorama, and the avatars. To fully utilize Augmentations, ArTekMed uses the SLAM reconstruction of the AR-HMD for occlusion handling in the real environment.**Human Perception within ArTekMed:** Occlusion handling with the SLAM reconstruction in AR allows users to understand the depth of virtual objects concerning real objects quickly. Stereoscopic rendering within the HMDs in AR and VR lets users observe the scene in 3D. Moreover, tracking of the HMD qualifies motion parallax as an additional visual cue for depth perception.**Dynamic UI in ArTekMed:** Conventional 2D in 3D UIs, such as menus and a radial menu to adjust settings and tools, allow users to interact with the system on a high level. Users interact with the ArTekMed system using their bodies and transform the point cloud reconstruction or digital avatar respectively for non-verbal communication. While interacting with the environment, diegetic virtual elements such as the 3D annotations and Magnorama are part of the digital world and fused with the real world in AR.**Evaluation in ArTekMed:** Novel systems such as the ArTekMed disrupt standard practices in Healthcare. Evaluation should therefore cover the fundamental acceptance of every component in the system and their usability. The evaluation covers the acceptance of teleconsultation versus conventional video calls [38], user representations [41], and advanced interaction techniques [42,43] - all with clinical use cases in mind.

ArTekMed combines digital sensors, AR/VR displays, and novel 3D user interfaces virtuously and intuitively. Furthermore, the augmentations in the form of 3D annotations in space and interactive aids for navigating and manipulating 3D space demonstrate the many-faceted synergy between the digital world, the user, and the interface.

### 3.2. Augmented Reality for Ophthalmology

In ophthalmic surgery, AR has the potential to optimize perception and understanding of the surgical scene. Conventionally, a surgical microscope provides visual access to the surgical scene, allowing visualization of the backside of the eye and an overhead view of the retina. Due to the fixed top view of the surgical site, depth is mainly inferred by shadows of surgical instruments projected onto the retina by a handheld light pipe. Today, multi-modal imaging has become available, providing complex multi-resolution imaging. In state-of-the-art setups, surgical sensing consists of 2D and 3D intraoperative Optical Coherence Tomography (iOCT) cross-sections along with the conventional microscopic view. This additional imaging provides high-resolution depth information of surgical instruments and anatomical structures and can be leveraged to extract semantic scene understanding and other helpful information for the surgeon. However, simultaneously providing all the available data to the surgeon may create high mental demand and visual overload, as surgeons must focus on various areas of multiple modalities, and mapping between these modalities is challenging.

Hence, there is a need for AR systems that can merge the data in an intelligent way by integrating the key elements of each modality and blending them such that a single image is generated that shows all the necessary information to the surgeon. This, in turn, will reduce the cognitive load on the surgeon and avoid distractions that occur when switching the gaze to look at other modalities. An example of a system fusing information from multiple sources in a fused overlay is the relevance-based visualization of X-Ray images in the camera space proposed by Pauly et al. [45].

Currently, there are two proposed approaches to support ophthalmic procedures: (i) visual AR, which directly augments the displayed imaging data to improve scene understanding and visual perception, and (ii) auditive AR, which leverages extracted information from data streams as parameters to generate or modify sound signals. The first work towards visual AR in ophthalmology was proposed in 2015 [46], in which the 2D iOCT cross-sections were augmented with the location of surgical tools, which the iOCT system cannot image due to signal attenuation. In this initial work, the augmented cross-sections were integrated next to the microscopic view in the surgical microscope. In an effort to reduce the visual overload, information that can be extracted from additional sensing, such as the predicted contact point between the instrument and retina from iOCT, can be visually augmented in the conventional microscopic view without integrating the complex imaging data [47]. On the other hand, advances in scanning speed paved the way toward visualizing surgeries only by means of real-time 3D iOCT, requiring new display paradigms. While GPU-accelerated volume rendering is capable of processing high data rates, additional perceptual cues have been shown to be a key to achieving an immersive and intuitive visualization that improves usability. In such an AR or VR system, both depth and distance perception need to be optimized and require a complete understanding of the surgical scene. In [48], the example of improving spatial perception through the use of coloring in volume rendering demonstrates that only in combination with semantic scene understanding can perceptual aids be employed in an optimal manner and in the most useful way for surgeons. Only with the combination of advanced sensing, data understanding, and perception concepts of AR can be effectively integrated into surgical setups and accepted by surgeons.

Moreover, auditive AR can also be an attractive way to convey supportive information in ophthalmic surgery while not overloading the surgical view with visual information. The works [16,22] related to auditive AR mentioned in Section 2.3 have been demonstrated on relevant examples in ophthalmology and show that sound can be used to improve situational awareness, but also to directly sonify parameters extracted from various imaging systems. These examples of applications related to eye surgery show that, especially for procedures that require delicate navigation and extreme dexterity, the correct perception is of utmost importance for the acceptance of both visual and auditive AR and requires a full semantic understanding of the surgical scene. Careful considerations need to be taken to select the type, amount, and channel of the augmentations in order not to overload but to effectively augment the surgical scene and, in the end, not to distract but to support the surgical task. Figure 5 shows an overview of Augmented Reality research projects developed within the CAMP chair for ophthalmologic procedures.

**Physical World in Ophthalmology:** The operating environment consists of a surgical microscope providing direct visual access to the ocular anatomy of the patient. The surgeon uses both hands to manipulate micro-surgical instruments and feet to control the microscope as well as the iOCT system via pedals on the floor. In modern setups, visualization of the operating area is provided by 3D monitors next to the patient site.**Sensors in Ophthalmology:** (Digital) operating microscopes providing a stereo view, intraoperative OCT lasers for 2D and 3D depth visualization.**Computer Perception in Ophthalmic Applications:** The operating microscope provides stereo RGB images, while 2D cross-sectional slices are acquired by the iOCT system. The compounding of these slices enables 3D, and in state-of-the-art systems, even temporal 4D visualization of the surgical area.**Digital World in Ophthalmic Applications:** The digital world consists of the raw imaging data of all sensors brought into a common coordinate frame, as well as the semantic information of anatomical structures and surgical instruments and information about their relationship. It can further contain information about the surgical phases.**Augmentations in Ophthalmic Applications:** Augmentations are either integrated into the surgical microscope or the 3D display or provided via audio signals and sonification methods. In both cases, they leverage semantic understanding provided by the digital world and aim to improve the perception of the surgical scene.**Human Perception in Ophthalmic Applications:** Depth and distance perception in iOCT volume renderings is mainly provided by color transfer functions and by generating sound signals or modifying musical pieces.**UI in Ophthalmic Applications:** The surgeon’s hands manipulate the light guide and tools. Access to the surgical microscope and iOCT system, hence, also the interaction with an AR system is mainly provided via foot switches. Automatic surgical phase understanding could improve the user interface for AR systems and reduce the cognitive load on the surgeon. The design of a user-centric UI is of utmost importance, allowing optimal usability of the system and optimal perception of the provided information without disturbing the surgeon’s workflow.**Evaluation in Ophthalmic Applications:** Ophthalmic applications are carefully evaluated on phantom eye models, in ex-vivo animal wet lab settings, and on surgical simulators. Close collaborations with surgeons during all stages of design, development, testing, and validation are required for ethical development. Only at a later stage in vivo animal studies will be conducted, and only then can systems be integrated into clinical studies on humans.

### 3.3. Camera-Augmented Mobile C-Arm

The motivation for proposing the Camera-Augmented Mobile C-arm (CAMC) [49,50] is to extend a mobile C-arm to provide surgical guidance via intuitive visualization. In the last two decades, we have followed the framework focusing on the different components of MAR, resulting in novel methodologies for X-Ray calibration, advanced AR visualizations through co-registration of RGB and X-Ray images, intra-operative navigation of surgical tools, 3D CBCT reconstruction, panoramic X-Ray image stitching, 3D-3D calibration for CBCT and RGBD data, radiation estimation and many more. An overview of CAMC systems is shown in Figure 6.

**Physical World in CAMC:** The main component of the physical world for CAMC is the deep-seated patient’s anatomy, which is sensed using a mobile X-ray system: Then, the visible patient surface, the surgical tools, the surgeon’s hand, and in later stages, the operating table, assistants, and the rest of the operating room. In the first versions of the CAMC, the co-registration of X-ray sensing and optical imaging improved surgical viewing. Later, the system got extended to include augmentation of full operating room interactions.**Computer Sensors and Perception in CAMC:** The sensors attached to the mobile C-arm have evolved over time. The first iteration started with a CCD camera near the X-ray source [49]. With the help of a double mirror system, it captures the live view of patient anatomy [51] fully registered with the X-ray view without any need for dynamic calibration. Later, the introduction of an RGB-D camera attached to the X-ray detector enabled acquiring a 3D representation of the surgical scene [52,53]. A recent advancement of CAMC adopted HMDs [54,55] both for visualization of the surgical site for the surgeon, as well as for tracking the C-arm.**Digital World in CAMC:** With each iteration of the system, CAMC creates different understandings of the digital world. However, the core aspect of the system is to associate a spatial-temporal relation of the imaging data provided by the C-arm, the patient anatomy captured by the attached cameras, the surgeon, and tools in the same coordinate frame [56].**Augmentations in CAMC Applications:** Patient’s pre-operative or intra-operative 2D and 3D medical data are augmented and fused with the live optical information of the patient. The system can further visualize the trajectory of tools and annotated points, lines, and planes.**Human Perception in CAMC Applications:** Machine Learning improves the perception of the scene by identifying relevant objects in both X-ray and optical images captured with the CAMC system to build a fused image for better handling occlusion [57]. In the presence of deformation, the best solution is to use intraoperative imaging such as ultrasound or optoacoustic imaging, and in particular cases, low-dose fluoroscopy to observe the motion of anatomical targets. Alternatively, scientists used endoscopic views of surface deformation and biomechanical models to estimate the deformation of the target anatomy for AR [58]. In addition, real-time tool tracking can provide the required precision for the execution of accurate surgical actions based on intraoperative computer-aided guidance [59]. With AR-HMDs, users can understand the spatial relationship between medical imaging data and patient anatomy more effortlessly.**UI in CAMC Applications:** With the HMD variant of the CAMC system, the user interaction is mostly through hand gestures and voice commands [60]. The user can manipulate the scale and position of the spatial X-ray images.**Evaluation in CAMC Applications:** The system was first evaluated on phantoms [51,61], then on human cadaver and ex-vivo animal anatomy [62,63] and finally through a set of patient studies [64,65,66]. Quantitative data such as radiation dose, planning and execution time, K-wire insertion accuracy, and other surgical tasks have been evaluated over the years with different types of the CAMC [55,63,67,68]. We also have performed qualitative evaluations of the system usability and user depth perception to help better understand and improve the visualization of CAMC.

### 3.4. Magic Mirror

Mirrors provide one of the most intuitive concepts for humans to understand the view from a different angle. Utilizing the mirror paradigm, we incorporate AR to facilitate the exploration of anatomy in the user’s augmented mirror reflection.

The *Magic Mirror* [69,70,71,72,73] combines a camera and a 2D monitor to create a view similar to conventional mirrors, as seen in Figure 7. The Magic Mirror recognizes the user’s pose with a camera. The Magic Mirror then computes the anatomical representation within its digital world accordingly. Rendering in situ augmentations of the digital anatomical representation within the camera’s color image gives the user the illusion of seeing their internal anatomy in addition to their typical mirror image. To further elaborate on the educational aspect of the Magic Mirror, it allows user interaction through arm gestures switching between different anatomical systems, or scrolling through transversal CT slices based on the height of the user’s right hand.

The Magic Mirror follows the principles of the MAR framework as follows:**Physical World of Magic Mirrors:** The physical world relevant to the Magic Mirror consists of its users. To complete the mirror view, the Magic Mirror additionally captures the environment behind the users.**Computer Sensors and Perception for Magic Mirrors:** An RGB-D camera facilitates both the mirror view and computes the pose of the users.**Digital World of Magic Mirrors:** The internal representation of the system consists of the user’s body pose and an anatomical model. The user’s pose deforms the virtual representation of their anatomy accordingly.**Rendering Displays of Magic Mirrors:** The Magic Mirror renders the image of its color camera on a large 2D monitor.**Augmentation of Magic Mirrors:** The virtual anatomies are augmented onto the color image. The system further enhances the illusion of looking into the body instead of seeing a simple overlay by utilizing a soft fall-off at the transition between the color image and the augmented view based on findings of Bichlmeier et al. [17].**Dynamic UI of Magic Mirrors:** As the anatomy augments the viewing users, they may use proprioception to move their hand to a specific organ with the visuo-proprioceptive feedback loop provided by the mirror and, once arrived, feel the location on their own body.**Evaluation of our Magic Mirror:** We evaluated the Magic Mirror with over 1000 medical students over multiple user studies, i.e., for Radiology [74] and Anatomy Learning [73]. Further fundamental research allowed us to understand the mirror paradigm for education [72].

The Magic Mirror highlights how AR can be utilized for education [74]. Ideally, users will be able to recall the learned and felt experience even in the absence of the visualization.

## 4. Conclusions

Decades of active research in AR and related areas were needed to enable the deployment of the first medical products using AR to support surgeons in the OR [65]. The current and upcoming MAR product releases are only a building block for future medical treatment alongside less invasive treatment methods, robot-assisted interventions, novel imaging modalities, and advanced reasoning using modern deep-learning techniques. MARS can be beneficial for smoothly integrating such new developments into the medical workflow and providing unique training opportunities.

Based on the experience we gathered during many years of active research and development, we propose a conceptual framework for MARS, emphasizing the necessity of adaptive interfaces and continuous evaluation with the target audience. Finally, we show the applicability of the proposed framework by modeling four MARS from our research group, which were all developed and evaluated in close collaboration with medical experts. We believe that building AR-based support systems for healthcare professionals requires an interdisciplinary approach, where state-of-the-art methods from computer vision, computer graphics, scene understanding, human-machine interaction, sensors, and displays need to be perfectly blended into a tailored experience that focuses on relevant information for the task and is intuitive to use. While we aim to model MARS specifically, many concepts and observations apply to AR systems in other domains.

Driven by the latest advances in machine learning, we expect to see more sophisticated methods for environment understanding, activity recognition, and surgical process tracking. These methods will need to be optimized to work in challenging healthcare delivery environments and to perform in real time so that developers and designers of MARS can build adaptive, context-aware user interfaces. Furthermore, we expect to see specialized solutions for distinct medical procedures that optimally combine the required sensing, display, processing, and interaction tailored to the needs of each stakeholder in the team.

This proposed framework summarizes the experiences we gained from years of creating MARS and teaching about the topic, and we hope that the community can learn from these insights and categorization and use it as a springboard to transform medical environments.

## Figures and Tables

**Figure 1 jimaging-09-00004-f001:**
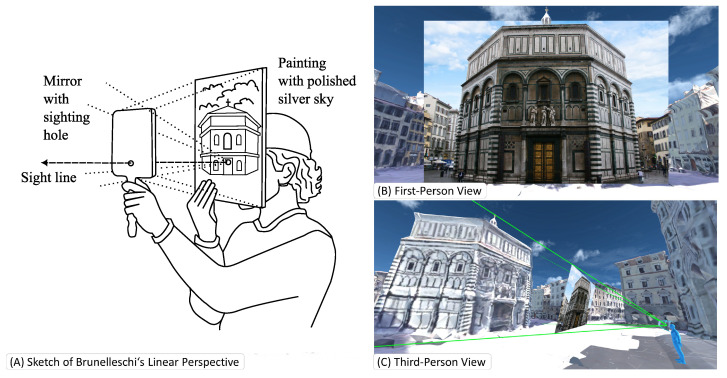
**The apparatus of Brunelleschi** consists of (**A**) a mirror, and a painted or printed image. Both components inhibit a hole for the user to look through. (**B**) The apparatus creates a linear projection that shows the image inside the user’s view. (**C**) The illustrated third-person view visualizes the frustum that is covered by the projection. Many modern AR applications use the same mathematical basis for creating in situ visualizations.

**Figure 2 jimaging-09-00004-f002:**
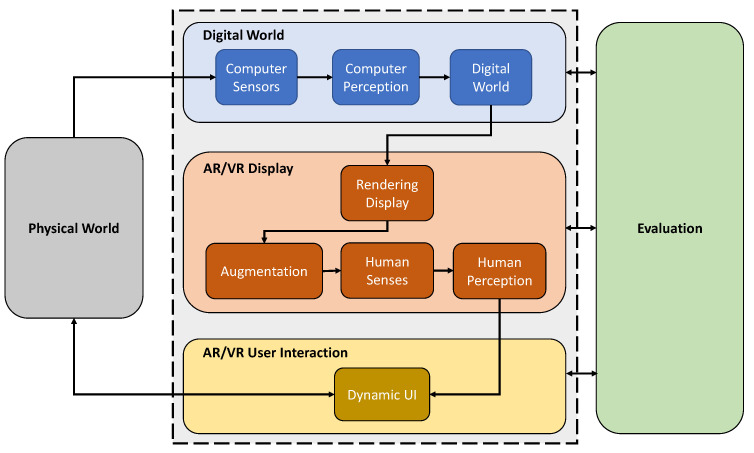
**The Medical Augmented Reality Framework** consists of four primary components: Digital World, AR/VR Display, AR/VR User Interaction, and evaluation. An MARS perceives the Physical World with its sensors and processes in a medium that users may perceive and interact with through the AR/VR interfaces. Evaluation is integral to the system’s conception, development, and deployment.

**Figure 3 jimaging-09-00004-f003:**
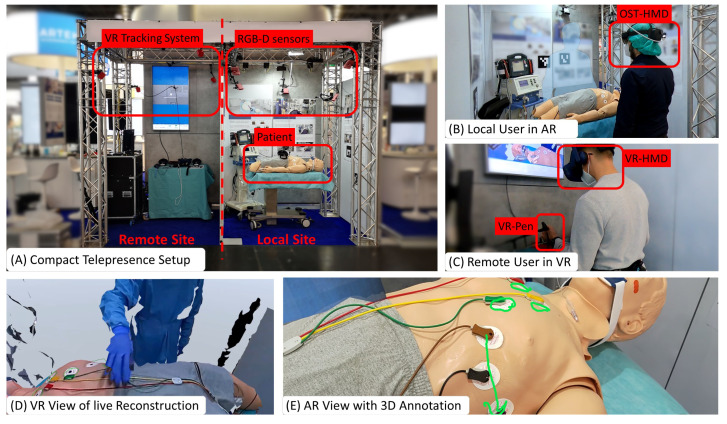
**Augmented Reality Teleconsultation System for Medicine** (ArTekMed) combines point cloud reconstruction with Augmented and Virtual Reality. (**A**) Capturing the local site with the patient requires extrinsically calibrated RGB-D sensors from which the system computes a real-time point cloud reconstruction. (**B**) The local user interacts with the real world while perceiving additional virtual content delivered with AR. (**C**) The remote user dons a VR headset and controller for interacting with the acquired point cloud. (**D**) The reconstruction represents the digital world known to the computer and is displayed to the VR User. (**E**) AR annotations made by the VR user is shown in situ on the patient.

**Figure 4 jimaging-09-00004-f004:**
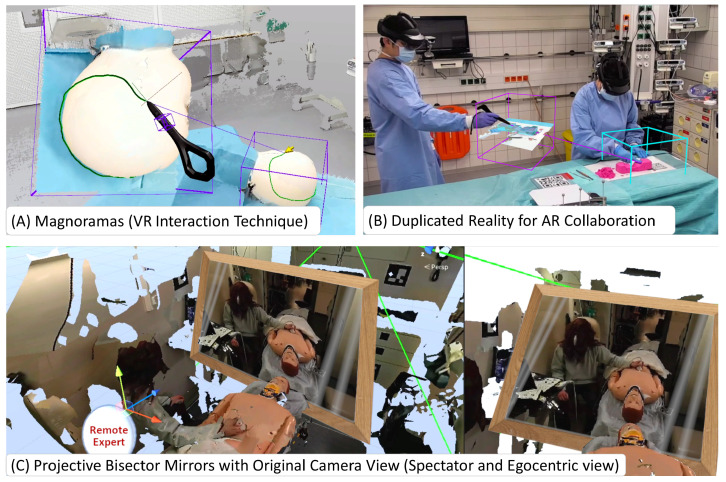
**Interaction Techniques Unique to ArTekMed**: (**A**) The Magnorama creates a dynamic 3D cutout from the real-time reconstruction and allows the user to interact and explore the space while intuitively creating annotations at the original region of interest within the duplicate. (**B**) The principle of Magnoramas translates well into AR. The resulting technique of Duplicated Reality allows co-located collaboration in tight spaces, even without a remote user. (**C**) For remote users to experience more details of the patient and their surroundings, ArTekMed deploys Projective Bisector Mirrors to bridge the gap between reality and reconstruction through the mirror metaphor.

**Figure 5 jimaging-09-00004-f005:**
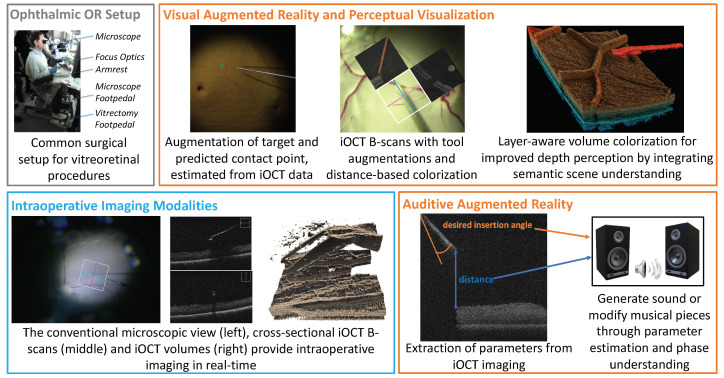
Typical setups in ophthalmic surgery consist of a complex operating area and multi-modal real-time imaging. Visual and auditive AR applications aim to improve perception and provide additional information while avoiding visual clutter and reducing the cognitive load of complex intraoperative data.

**Figure 6 jimaging-09-00004-f006:**
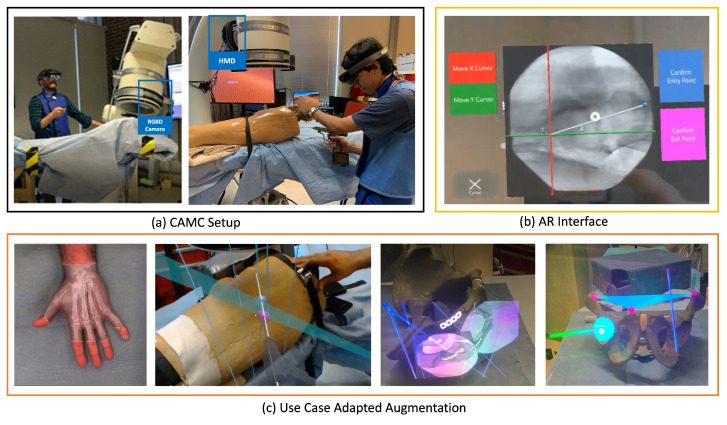
**CAMC** aims to reduce the need for ionizing radiations and to provide spatially aware, intuitive visualization of joint optical and fluoroscopic data. (**a**) Calibration of the C-arm with the patient and the technician and surgeon’s HMD enables efficient surgical procedures in a collaborative ecosystem. (**b**) Advanced AR interface aids in better planning trajectories on the X-ray acquisitions. (**c**) The adaptive UI and augmentations in intra-operative planning and execution support various image-guided procedures.

**Figure 7 jimaging-09-00004-f007:**
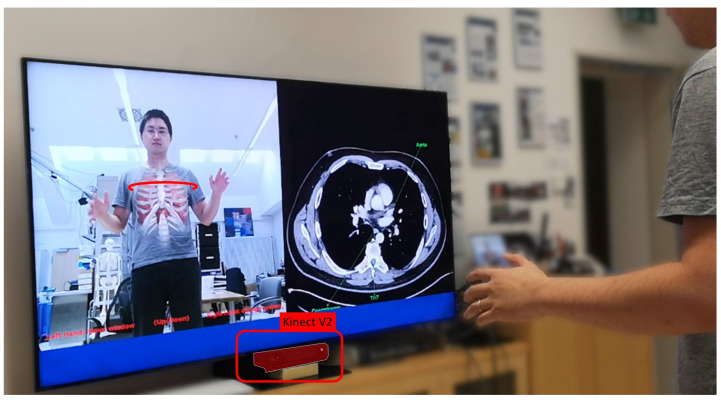
**The Magic Mirror** visualizes anatomical structures in situ on the *mirror* reflection of the user in front of the RGB-D camera. Additionally, our Magic Mirror system displays transverse slices of a CT volume on the right half of the monitor that matches the slice selected by the user with their right hand.

## Data Availability

Not applicable.

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
