# Peer review of "Medical Augmented Reality: Definition, Principle Components, Domain Modeling, and Design-Development-Validation Process"

_2313-433X, 2022, doi:10.3390/jimaging9010004_

Round 1

Reviewer 1 Report

The authors provide an extensive review of MARS which includes their own extensive experience. The manuscript could be shortened and much of the historical citations removed. The authors often make statements about aspects of the system without substantiating these with data from their lab. For example, on page 6 they state: "when the users of a Medical AR system observe the internal organs not usually visible to their eyes or hear the augmented sound of an organ touched by a surgical instrument, the experience needs to be perceptually consistent, allowing the users to recognize well and localize the objects otherwise invisible to the users’ unaided senses." Can the authors provide some data that the users actually do recognize the objects. 

As another example they state on page 8: The users are the focus of all AR systems and should be considered accordingly during the conception and development of AR solutions." While this intuitively makes sense, it would help the reader if the authors provide specific examples where an investigator performed assessments that aided in the iteration of a MARS leading to its commercial introduction. 

On page 9 the authors state: "We therefore highly recommended thinking of alternative ways for quantitative measurements of human performance, indicating the success of mental mapping or measuring fatigue and/or dissatisfaction." Fatigue measured how and compared to what. A randomized trial of task performance without MARS vs. with MARS?

On page 12: "Hence, there is a need for AR systems that augment the surgical scene and transfer information from multi-modal data streams without distracting the surgeon’s gaze from the operating area." The eye/brain does well processing one image and maybe two if they are close and have related content. Sending multiple images to the user who may be focused on a task (surgery) would not likely be processable. The authors should present data demonstrating that it is possible to manage multiple feeds and that these don't lead to distraction.  

Author Response

The authors provide an extensive review of MARS which includes their own extensive experience. The manuscript could be shortened and much of the historical citations removed. 

We agree that parts of the manuscript could be shortened, therefore we shortened parts of section 3. However, we would like to keep the historical background of Augmented Reality, as we consider the work of Brunelleschi to be the first Augmented Reality system in the world.

The authors often make statements about aspects of the system without substantiating these with data from their lab. For example, on page 6 they state: "when the users of a Medical AR system observe the internal organs not usually visible to their eyes or hear the augmented sound of an organ touched by a surgical instrument, the experience needs to be perceptually consistent, allowing the users to recognize well and localize the objects otherwise invisible to the users’ unaided senses." Can the authors provide some data that the users actually do recognize the objects. 

We agree that we did not back the claims, if users could recognize the AR objects. Therefore we added two examples of experimental evidence right after the statement, that users could recognize the AR visualizations, and that they are useful in a surgical setting. We added an example in both auditive AR (Matinfar et al.) and visual AR (Okur et al.). This is the added paragraph:

“Evidence that they can, in fact, integrate the additional information and are affected by the augmented visuals and sounds can be explored in experiments with users: For example, Okur et al. analyzed the surgeons' visualizations preference (either AR or VR) in intra-operative SPECT acquisitions. The surgeons relied on both AR and VR visualizations and stated that the two modes complemented each other. In an auditive AR experiment, Matinfar et al. had users recognize relevant sensor changes in retinal surgery. The users could accurately and quickly recognize the position of the tool.”

As another example they state on page 8: The users are the focus of all AR systems and should be considered accordingly during the conception and development of AR solutions." While this intuitively makes sense, it would help the reader if the authors provide specific examples where an investigator performed assessments that aided in the iteration of a MARS leading to its commercial introduction

We agree that we did not sufficiently back up this statement in the previous version of the manuscript and have added the following sentence in the respective section:

“One example for the iterative inclusion of the user towards the development of a successful system is a project conducted within our lab that integrated VR and AR for intraoperative single-photon emission computerized tomography (SPECT) which was started in 2007 [1] and resulted in the first MARS used in surgery with CE certification and FDA clearance in 2011 [2].”

[1] Towards intra-operative 3D nuclear imaging: reconstruction of 3D radioactive distributions using tracked gamma probes, T Wendler, A Hartl, T Lasser, J Traub, F Daghighian, SI Ziegler, N Navab, International Conference on Medical Image Computing and Computer-Assisted Intervention, 2007

[2] MR in OR: First analysis of AR/VR visualization in 100 intra-operative Freehand SPECT acquisitions, A Okur, SA Ahmadi, A Bigdelou, T Wendler, N Navab, 2011 10th IEEE International Symposium on Mixed and Augmented Reality, 211-218, 2011

On page 9 the authors state: "We therefore highly recommended thinking of alternative ways for quantitative measurements of human performance, indicating the success of mental mapping or measuring fatigue and/or dissatisfaction." Fatigue measured how and compared to what. A randomized trial of task performance without MARS vs. with MARS?

Thank you for the comment and questions. Fatigue can be qualitatively measured with NASA TLX questionnaire as it is part of the workload evaluation. In NASA TLX, “mental demand” and “physical demand” are two indicators for both mental and physical fatigue whereas in SurgTLX  “physical fatigue” is being measured.

To further clarify this statement, we have added the following sentence to the respective section of the manuscript: 

“To provide a more objective measurement of computational systems on humans, the community of affective computing has been actively creating and evaluating new ways of quantifying human affects. By applying a learning-based algorithm, multimodal digital data that is collected by wearable sensors can report and inform the physical and mental fatigue levels [3].”

[3] Luo, H.; Lee, P.A.; Clay, I.; Jaggi, M.; De Luca, V. Assessment of fatigue using wearable sensors: a pilot study. Digital biomarkers 2020, 4, 59–72

On page 12: "Hence, there is a need for AR systems that augment the surgical scene and transfer information from multi-modal data streams without distracting the surgeon’s gaze from the operating area." The eye/brain does well processing one image and maybe two if they are close and have related content. Sending multiple images to the user who may be focused on a task (surgery) would not likely be processable. The authors should present data demonstrating that it is possible to manage multiple feeds and that these don't lead to distraction.  

Thank you for pointing this out, we agree that we failed to sufficiently explain this point in the manuscript. The goal of MARS is to achieve a perceptual fusion from multiple data sources and not having multiple parallel images / information that can be distracting. We have added the following sentence to the respective section of the manuscript: 

“Various streams of multimodal data and information can lead to visual clutter and cognitive overload when the data is displayed side-by-side. Hence, there is a need for AR systems that can merge the data in an intelligent way by integrating the key elements of each modality and blend them together, such that a single image is generated that shows all necessary information to the surgeon. This in turn will reduce the cognitive load on the surgeon and avoid distractions that occur when switching the gaze to look at other modalities. An example for a system fusing information from multiple sources in a fused overlay is the relevance-based visualization of X-Ray images in the camera space proposed by Pauly et al. [4].”

[4] Supervised classification for customized intraoperative augmented reality visualization, Pauly, Olivier, Katouzian, Amin, Eslami, Abouzar, Fallavollita, Pascal, and Navab, Nassir, In Proceedings of 2012 IEEE International Symposium on Mixed and Augmented Reality (ISMAR), pp 311-312, Atlanta, GA, USA, 2012.

Reviewer 2 Report

Very nice description of the Brunelleschi's concept using both Youtube video for visualization and the framework of Augmented Reality in Figure 2.  

Section 3 on applications of AR is very elaborate and detailed, some of it can be condensed and made more focused and brief.

The Font size in the inset of Figure 5 can be enlarged a bit to make it easier for the reader.

Section 4 on Conclusion is very crisp and coherent. Overall great effort by the authors.

Author Response

Very nice description of the Brunelleschi's concept using both Youtube video for visualization and the framework of Augmented Reality in Figure 2.  

Section 3 on applications of AR is very elaborate and detailed, some of it can be condensed and made more focused and brief.

Thank you for the very valid comment, we have shortened the general project descriptions of the projects “AR for ophthalmology” and “ARTEKMED” which were extensive in the previous version of the manuscript.

The Font size in the inset of Figure 5 can be enlarged a bit to make it easier for the reader.

Thank you for pointing this out, we have changed the figure accordingly.

Section 4 on Conclusion is very crisp and coherent. Overall great effort by the authors.

Reviewer 3 Report

The manuscript is about Medical Augmented Reality, more specific the definition, principle components, domain modeling, and the design-development-validation process. In summary, for someone who works since years in Medical Augmented Reality, the manuscript is very interesting and well-written, as it provides a good overview over recent developments and challenges in the field. It should be pointed out that while the insights are extensive, they are almost exclusively based on the findings of the research group of the authors. As such, I am not sure if the contribution fits the category or is within the scope of a (Research) “Article”, since it presents no novel problems or solutions, and for a review article, it is not comprehensive enough. I recently read a “Perspective” about a Healthcare Metaverse, which would be a better fitting category (the manuscript would also fit an “Expert Option”):
https://www.nature.com/articles/s42256-022-00549-6

The reader should be pointed to two recent surveys in Medical Augmented Reality for further reading:
https://doi.org/10.1016/j.media.2022.102361
https://arxiv.org/abs/2209.03245

Otherwise, I have no further comments and leave it to the editor if the manuscript fits the scope of the journal as “Article”.

Author Response

The manuscript is about Medical Augmented Reality, more specific the definition, principle components, domain modeling, and the design-development-validation process. In summary, for someone who works since years in Medical Augmented Reality, the manuscript is very interesting and well-written, as it provides a good overview over recent developments and challenges in the field. It should be pointed out that while the insights are extensive, they are almost exclusively based on the findings of the research group of the authors. As such, I am not sure if the contribution fits the category or is within the scope of a (Research) “Article”, since it presents no novel problems or solutions, and for a review article, it is not comprehensive enough. I recently read a “Perspective” about a Healthcare Metaverse, which would be a better fitting category (the manuscript would also fit an “Expert Option”):

https://www.nature.com/articles/s42256-022-00549-6

We agree with the reviewer and have contacted the editors separately to change the manuscript type to “Concept Paper” which fits the content of the paper perfectly in our opinion.

The reader should be pointed to two recent surveys in Medical Augmented Reality for further reading:

https://doi.org/10.1016/j.media.2022.102361

https://arxiv.org/abs/2209.03245 

Thank you for the valuable feedback, we have added the following sentence in the introduction section:

“We furthermore point the reader to the literature review about Medical Augmented Reality by Sielhorst et al. from 2008 [5] and a more recent review about the use of head mounted displays in surgery by Birlo et al. [6]. Furthermore, an example for the adoption of one of the most used commercially available HMD systems is given in a review paper by Gsaxner et al. [7].”

[5] Sielhorst, T.; Feuerstein, M.; Navab, N. Advanced Medical Displays: A Literature Review of Augmented Reality. Journal of Display Technology 2008, 4, 451–467.  https://doi.org/10.1109/JDT.2008.2001575.

[6] Birlo, M.; Edwards, P.E.; Clarkson, M.; Stoyanov, D. Utility of optical see-through head mounted displays in augmented reality-assisted surgery: A systematic review. Medical Image Analysis 2022, 77, 102361. https://doi.org/https://doi.org/10.1016/j.media.2022.102361.

[7] The HoloLens in Medicine: A systematic Review and Taxonomy. arXiv:2209.03245 2022.

Otherwise, I have no further comments and leave it to the editor if the manuscript fits the scope of the journal as “Article”.

Round 2

Reviewer 1 Report

Thank you for the revisions. I understand your desire to want to retain the information and history on Brunelleschi but this section should be trimmed as it is a distraction from the rest of the manuscript. 

All of the explanations are sufficient except the last one: "The goal of MARS is to achieve a perceptual fusion from multiple data sources and not having multiple parallel images / information that can be distracting" While the citation by Pauly (4) helps, fusing an archived image (static fluoroscopy) into the real world (in this case an overlay) can't compensate for organ movement. This is one of the issues that companies involved with 3D navigation need to manage. Further description of how this problem can be addressed is warranted. 

Author Response

Thank you for the revisions. I understand your desire to want to retain the information and history on Brunelleschi but this section should be trimmed as it is a distraction from the rest of the manuscript. 

We agree with your feedback and have shortened the introduction accordingly.

All of the explanations are sufficient except the last one: "The goal of MARS is to achieve a perceptual fusion from multiple data sources and not having multiple parallel images / information that can be distracting" While the citation by Pauly (4) helps, fusing an archived image (static fluoroscopy) into the real world (in this case an overlay) can't compensate for organ movement. This is one of the issues that companies involved with 3D navigation need to manage. Further description of how this problem can be addressed is warranted. 

Thank you for the constructive feedback, we have added the following sentences to the manuscript to explain how to account for tissue deformation and moving anatomy in MARS.

“In the presence of deformation, the best solution is to use intraoperative imaging such as ultrasound or optoacoustic imaging, and in particular cases, low-dose fluoroscopy to observe the motion of anatomical targets. Alternatively, scientists used endoscopic views of surface deformation and biomechanical models to estimate the deformation of the target anatomy for AR [1]. In addition, real-time tool tracking can provide the required precision for the execution of accurate surgical actions based on intraoperative computer-aided guidance [2].”

[1] Paulus, C.J.; Haouchine, N.; Cazier, D.; Cotin, S. Augmented Reality during Cutting and Tearing of Deformable Objects. In Proceedings of the 2015 IEEE International Symposium on Mixed and Augmented Reality, 2015, pp. 54–59. https://doi.org/10.1109/ISMAR.2015.19.

[2] Pakhomov, D.; Premachandran, V.; Allan, M.; Azizian, M.; Navab, N. Deep Residual Learning for Instrument Segmentation in Robotic Surgery. In Proceedings of the Machine Learning in Medical Imaging; Suk, H.I.; Liu, M.; Yan, P.; Lian, C., Eds.; Springer International Publishing: Cham, 2019; pp. 566–573.
